# Revisiting the MMTV Zoonotic Hypothesis to Account for Geographic Variation in Breast Cancer Incidence

**DOI:** 10.3390/v14030559

**Published:** 2022-03-09

**Authors:** Alexandre F. R. Stewart, Hsiao-Huei Chen

**Affiliations:** 1Laboratory of Translational Genomics, Ruddy Canadian Cardiovascular Genetics Centre, University of Ottawa Heart Institute, Ottawa, ON K1Y 4W7, Canada; 2Department of Biochemistry, Microbiology and Immunology, University of Ottawa, Ottawa, ON K1H 8M5, Canada; 3Centre for Infection, Immunity and Inflammation, University of Ottawa, Ottawa, ON K1H 8M5, Canada; 4Department of Medicine, University of Ottawa, Ottawa, ON K1H 8M5, Canada; 5Department of Cellular and Molecular Medicine, University of Ottawa, Ottawa, ON K1H 8M5, Canada; 6Brain and Mind Institute, Ottawa Hospital Research Institute, Ottawa, ON K1H 8M5, Canada; 7Neuroscience Division, Ottawa Hospital Research Institute, Ottawa, ON K1H 8M5, Canada

**Keywords:** breast cancer, geographic variation, house mice, MMTV, zoonosis

## Abstract

Human breast cancer incidence varies by geographic location. More than 20 years ago, we proposed that zoonotic transmission of the mouse mammary tumor virus (MMTV) from the western European house mouse, *Mus musculus domesticus*, might account for the regional differences in breast cancer incidence. In the intervening years, several developments provide additional support for this hypothesis, including the limited impact of genetic factors for breast cancer susceptibility revealed by genome-wide association studies and the strong effect of antiretroviral therapy to reduce breast cancer incidence. At the same time, economic globalization has further expanded the distribution of *M. m. domesticus* to Asia, leading to a significant increase in breast cancer incidence in this region. Here, we revisit this evidence and provide an update to the MMTV zoonotic hypothesis for human breast cancer at a time when the world is recovering from the global COVID-19 zoonotic pandemic. We present evidence that mouse population outbreaks are correlated with spikes in breast cancer incidence in Australia and New Zealand and that globalization has increased the range of *M. m. domesticus* and MMTV. Given the success of global vaccination campaigns for HPV to eradicate cervical cancer, a similar strategy for MMTV may be warranted. Until breast cancer incidence is reduced by such an approach, zoonotic transmission of MMTV from mice to humans as an etiologic factor for breast cancer will remain controversial.

## 1. Introduction

Environmental factors play an important part in the etiology of sporadic breast cancer [1]. Geographic variation in human breast cancer incidence is well-recognized, and the country of residence accounts for ~80% of the lifetime risk [2]. Moreover, for individuals who migrate from lands where the incidence is low to where the incidence is high, their breast cancer incidence gradually increases up to 2–3-fold over several decades after immigration [3,4,5]. Intriguingly, neither exposure to organochlorine pesticides [1,6], nor reduced phytoestrogen intake or other changes in diet [7], nor increased cigarette smoking or alcohol intake [8] account for this increase in breast cancer incidence. Thus, environmental factors endemic to regions of high breast cancer incidence remain to be identified. In contrast to the profound geographic effect, the genetic contribution is relatively minor. While up to 20% of breast cancer risk occurs in families [9,10], families tend to remain in specific geographic areas and share environmental as well as genetic factors. Since many studies are conducted in the United States, a country with one of the highest incidences of breast cancer, the genetic contribution to breast cancer risk may be overestimated (and include the geographic environmental factor). Genome-wide association studies to identify common genetic variants associated with the risk of breast cancer have relied predominantly on cohorts residing in the United States, Canada, Australia and Western Europe, regions with high breast cancer incidence [11,12]. For the 77 most common genetic variants for breast cancer, women carrying most of these variants had a lifetime risk of breast cancer increased by less than 8% [13], highlighting the limited contribution of genetic risk to breast cancer.

Could the environmental factor be a cancer-inducing virus? If so, it would need to have very low infectivity to maintain the geographic differences in incidence. A highly infectious virus would spread rapidly and eliminate geographic differences in breast cancer incidence. For example, human papillomavirus-induced cervical cancer shows geographic variation, with the highest incidence in sub-Saharan Africa and Melanesia (http://gco.iarc.fr/today, accessed on 8 March 2022). Infectivity is predominantly by sexual contact, but is relatively low, with 20% transmission in the first 6 months between heterosexual couples [14]. Viral-induced oncogenic transformation takes time, such that cervical cancer is typically diagnosed in women between the ages of 35–45. Nonetheless, HPV is now proven to cause cervical cancer, since the incidence of cervical cancer has been eliminated in women born after 1995 who are vaccinated against HPV [15]. 

In the case of breast cancer, while a human oncogenic breast cancer virus has not been identified, similar low infectivity and a slowly transforming viral pathogen might account for the slow increase in breast cancer incidence in migrants, requiring years to become manifest. The most plausible environmental factor that could account for geographic variation in human breast cancer incidence is the zoonotic transmission of the mouse mammary tumor virus (MMTV) from mice to humans, given the geographic distribution of mouse strains carrying different loads of MMTV [16]. MMTV is a betaretrovirus that causes mammary tumors in mice [17] and is detected in up to 40% of human breast tumors [18]. Retroviruses are intimately adapted to their hosts, such that transmission across species is limited. In the case of MMTV, the virus transmitted in breast milk infects pups through lymphocytes at Peyer’s patches in the duodenum [19], and the superantigen response and infected lymphocytes ensure MMTV is transmitted to mammary glands that are formed in the first postnatal weeks [20]. Mice reach sexual maturity at 4 weeks of age [21], ready to begin the cycle of transmission anew. Mammary tumors do not appear until 8–18 months, depending on environmental stresses [22]. In contrast, human females reach puberty at ~12–14 years of age, and breast cancer does not appear until many years later, with the risk increasing with a later age of menopause onset [23]. Given these chronological differences, the acquisition of MMTV by humans is likely to be infrequent and limited. Even though MMTV has been detected in dental calculus from ancient human skulls [24], suggesting its presence in saliva, MMTV is unlikely to be transmitted easily between humans, given the presence of different viral strains in a family cluster [25]. 

In contrast to other tumor types that display varying levels of immunogenicity (where the host recognizes and raises an immune response against the tumors) [26], MMTV-positive mammary tumors in mice are poorly immunogenic, highly tumorigenic, invasive, and spontaneously metastasize to distant organs [27]. Early observations on the poorly immunogenic properties of human breast tumors [28] led Stewart et al. to examine the consequence of chronic immunosuppression on breast cancer in women chronically immunosuppressed after organ transplantation [29]. Remarkably, these women have a reduced incidence of de novo breast cancer, suggesting that the weak immune response promotes human breast cancer growth, as in mice [29]. In the same year, Beatriz Pogo’s group identified MMTV-like sequences in human breast tumors [18]. 

We proposed that MMTV might be transmitted to humans from house mice, particularly from the Western European house mouse *Mus musculus domesticus*, and account for the geographic variation in breast cancer incidence [16]. House mice are a commensal species; they live where humans live and eat what humans eat. Since we proposed this zoonotic hypothesis for MMTV contributing to human breast cancer, there have been many reports confirming the presence of MMTV in a subset of human breast tumors (see the meta-analysis by Wang et al. [30] and extensive references therein). Our hypothesis for MMTV zoonosis to account for geographic differences in breast cancer incidence made several assumptions: (1) Different species of house mice inhabit and are established in different regions and this distribution has remained constant over time. (2) Different species of house mice shed different strains or different viral loads of exogenous MMTV. (3) Susceptibility to MMTV infection would be similar among different human populations, explaining the migrant effect. (4) Mouse population densities will be correlated with viral transmission to humans. Here, we re-examined the evidence to support or refute these assumptions.

## 2. Materials and Methods

### 2.1. Testing for Association of Breast Cancer and M. m. domesticus Populations

Female breast cancer world age-standardized incidence rates (WASIR) were compared by geographic location for data available at the time of our previous report [16] to data available currently from the International Agency for Cancer Research of the World Health Organization (https://gco.iarc.fr/, accessed on 6 December 2021). For countries in Europe, data were binned according to lands of *M. m. domesticus*, *M. m. musculus* and lands in between where hybrid mice are found based on genotyping and phylogenetic evidence; and mean WASIR were compared by two-way random measures ANOVA to determine whether they differ by mouse populations, whether they have changed over time, and whether there is an interaction between mouse populations and time. Post-hoc comparisons were carried out using Sidak’s test and differences were considered significant at the *p* < 0.05 level, after correcting for multiple comparisons. For non-European locations, WASIR were similarly compared between countries or localities where *M. m. domesticus* is the prevalent species to countries or localities where other mice (*M. m. musculus*, *M. m. castaneus*) are the resident species. Statistical analysis was carried out using GraphPad Prism software.

### 2.2. Correlation of Mouse Population Outbreaks and Annual Breast Cancer Incidence Rates

To address the hypothesis that mouse population density might correlate with viral load and human exposure to MMTC, the annual incidence rates of breast cancer for New South Wales, Australia (https://www.cancer.nsw.gov.au/, accessed on 3 January 2022) and New Zealand (https://www.health.govt.nz/publication/cancer-historical-summary-1948–2017, accessed on 3 January 2022) were plotted between 1972 and 2001. In these two localities, where *M. m. domesticus* is the endemic mouse species, mouse population outbreaks occur with cyclical regularity and have been well documented [31,32,33]. 

## 3. Results

### 3.1. Breast Cancer Incidence Rates Still Associate with M. m. domesticus Range in Europe

There are essentially 3 sub-species of the house mouse *Mus musculus* (*M. m.*): *M. m. musculus* found in eastern Europe and Asia, *M. m. castaneus* found in Asia, and *M. m. domesticus*, originally from the Middle East. They are thought to have spread to North Africa and Western Europe, but whose range spread globally during the period of European colonization to North and South America, Hawaii, Australia, New Zealand and parts of sub-Saharan Africa [34,35,36]. In European countries segregated by their endogenous mouse populations, breast cancer incidence rates continue to show a strong association with the range of *M. m. domesticus* (Table 1 and Figure 1A). Two-way random measures ANOVA showed a significant main effect of location (F_2,23_ = 20.61, *p* < 0.0001) and time (F_1,23_ = 114.3, *p* < 0.0001), reflecting a significant increase in the breast cancer incidence rate overall, with a nearly 1.7-fold increase in Eastern Europe. There was no interaction between the incidence rate by mouse location and time, suggesting that the range of *M. m. domesticus* has remained relatively constant. Pairwise comparisons between the mean rates for *M. m. domesticus* lands and lands with hybrid mice, were not significant but were significant between *M. m. domesticus* and *M. m. musculus* lands (*p* < 0.0001 for both 1997 and 2020) and between the hybrid lands and *M. m. musculus* lands (*p* = 0.0005 for 1997 and *p* = 0.0036 for 2020). This result suggests that hybrid mice carry the same risk of transmitting MMTV to humans as do *M. m. domesticus* mice.

### 3.2. Breast Cancer Incidence Rates Are No Longer Segregated by M. m. domesticus Range outside of Europe

In non-European countries (excluding sub-Saharan Africa), the incidence of breast cancer has remained elevated in lands where *M. m. domesticus* is the resident or introduced species (Table 2). In contrast, lands traditionally considered to be primarily populated by *M. m. musculus* or *M. m. castaneus* have seen a nearly 3-fold increase in breast cancer incidence rates, such that the difference in breast cancer incidence between lands of *M. m. domesticus* and these other mice is no longer significant (Table 2 and Figure 1B). Two-way random measures ANOVA showed a significant effect of location (F_1,21_ = 8.314, *p* = 0.0089), of time (F_1,21_ = 25.13, *p* < 0.0001) and a significant interaction between location and time (F_1,21_ = 4.617, *p* = 0.0435), indicating a change over time by location. Pairwise comparisons showed a significant association of breast cancer incidence rates with lands of *M. m. domesticus* in 1997 (*p* = 0.0021) but no longer in 2020 (*p* = 0.1745). At face value, this result undermines the zoonotic hypothesis if one assumes the ranges of these mice have not changed or that *M. m. domesticus* has not hybridized with *M. m. musculus* or *M. m. castaneus* mice in these locations. 

### 3.3. Cyclical Mouse Population Outbreaks Precede Increases in Breast Cancer Incidence Rates

Outbreaks in mouse populations have been documented in Australia [31,32], New Zealand [33], Hawaii [37], and California [38]. If the mouse population density bursts in the vicinity of urban centers, and increases the viral exposure of humans, one would expect transient increases in breast cancer incidence rates after a certain delay, following increased exposure. In mice, mammary tumors in the susceptible C3H/He strain develop between 8 and 18 months, dependent on environmental stress, which greatly accelerates tumor appearance [22]. We do not know how long after a mouse population outbreak exposed humans display an increased breast cancer incidence if mice from rural areas invade urban centers. In New South Wales, Australia, the cancer registry documented a marked increase in breast cancer incidence rates from 1972 to 2001, and the increase was not linear but displayed cyclical increases followed by temporary drops (Figure 2A). New South Wales, one of the principal wheat-growing regions of Australia, experiences cyclical outbreaks in the *M. m. domesticus* mouse population [31,32]. Even in 2021, this problem was ongoing and severe, as highlighted in this video (https://www.youtube.com/watch?v=sJysxVeVusU, accessed on 8 March 2022). The plotting of the mouse outbreaks on a chart of breast cancer incidence rates suggests that mouse outbreaks contribute to a modest increase in breast cancer incidence rates, and have a lag of approximately 3 years. For New South Wales, after the 1979 outbreak, the annual breast cancer incidence rate increased 15 times in the following 22 years (until 2001), resulting in a probability of 0.682, but occurred 7 out of 9 times within 3 years after an outbreak (the probability that this occurred by chance is *p* = 0.069). Similarly for New Zealand, mouse population outbreaks in the Orongorongo Valley, located about 40 km from Wellington, have been documented over 25 years from 1972 to 1996 and trap densities of greater than 10 per 100 traps/night [33] were included in the chart (Figure 2B). Here too, a lag of approximately 3 years follows mouse population outbreaks. The first outbreak for New Zealand was recorded in 1976 and breast cancer incidence rates increased 13 times in the next 25 years (probability of 0.52) but within 3 years of each of the 5 population outbreaks (the probability that this occurred by chance is *p* = 0.038). As a control, we included age-standardized incidence rates for all of Canada, covering a vast geographic area (Figure 2C). Mouse population outbreaks are not well documented in Canada but would affect regional rather than national incidence rates. Of note, a deer mouse (*Peromyscus maniculatus*) population surge in 1985 was noted in the forests on Ontario [39], 3 years before a surge in breast cancer incidence (Ontario is Canada’s largest province by population). Otherwise, breast cancer incidence was relatively steady from 1988 to 2001.

## 4. Discussion

Here, we have revisited the MMTV zoonosis hypothesis that we proposed 22 years ago to account for the geographic variation in human breast cancer incidence rates [16]. In addition, we compared the incidence rates of breast cancer in Australia and New Zealand where mouse population outbreaks have been well documented over a long period of time to correlate mouse densities to changes in human breast cancer incidence rates. Consistent with our prior study, incidence rates remained associated with the range of *M. m. domesticus* in Europe, but this association no longer held for other countries outside of Europe, likely because *M. m. domesticus* range has expanded to countries where it was not present before (see below). 

Does globalization account for the redistribution of *M. m. domesticus* and MMTV?

Geographic variation in breast cancer incidence has been reported within China, with a higher incidence in northern China correlated with the presence of MMTV-like DNA sequences detected in breast tumors; in 22% of breast tumor specimens from Hebei (northern China) and only 5.7% of specimens from Jiande (southern China) [30]. Furthermore, a meta-analysis of published studies identifying MMTV-like DNA sequences from various locations confirmed the correlation of MMTV-like viral sequences with the *M. m. domesticus* distribution [30]. The highest breast cancer incidence in continental China was reported for Shanghai, where the age-standardized incidence rate from 1973 to 2012 rose from 17/100,000 to 41/100,000 [41]. It is worth noting that Shanghai is the largest harbor in the world (https://www.worldshipping.org/top-50-ports, accessed on 8 March 2022) and that a country-wide survey of mice in China, classifying mice using mitochondrial D-loop sequences, confirmed the presence of wild-caught *M. m. domesticus* mice in Shanghai [42]. This finding suggests that China is no longer a land strictly of *M. m. musculus* and *M. m. castaneus*. Furthermore, given the ability of *M. m. domesticus* mice to hybridize with *M. m. castaneus* mice introduced to North America at lake Casitas in California [43], the introduction of *M. m. domesticus* to China is likely to have a significant impact on breast cancer incidence in the coming years.

Taiwan is another example where the incidence has gone from 17/100,000 in 1995 [44] to 93 per 100,000 in 2017 [45], as high as in the United States. While *M. m. domesticus* mice have not been identified in Taiwan, *M. m. castaneus* mice were found to have some *M. m. domesticus* haplotypes in mitochondrial D-loop sequences [46], perhaps reflecting a historical admixture from *M. m. domesticus* mice introduced by the Portuguese in the 16th century, or the Spaniards in the 17th century, or more recently from wheat shipments from the United States, Australia and Canada (https://www.statista.com/statistics/1058261/taiwan-import-market-share-of-durum-wheat-by-country/, accessed on 8 March 2022), including all lands of *M. m. domesticus*. It is worth noting that the incidence of murine typhus, transmitted to humans from mice by fleas infected with *Ricketsia typhi*, is highest in the proximity of major international seaports in Taiwan [47]. While pet dog ownership in Taiwan and China has become very popular in recent years [48], and while an initial retrospective study suggested that dog owners were at increased risk of developing breast cancer [49], this observation has not been confirmed in a larger prospective study [50]. As we suggested previously, direct fecal-oral transmission from mice to humans seems more likely [16].

The consumption of ultra-processed foods with increasing wealth might be a confounding factor contributing to the increased incidence of breast cancer worldwide [51]. However, it is important to note that while Europe has seen an increase in breast cancer incidence overall, processed food consumption in women has fallen in Austria, Belgium, Denmark, the Netherlands, Sweden [52]; countries where incidence increased or remained elevated. Thus, while increased processed food consumption in non-European countries might have contributed to increased breast cancer incidence, it cannot account for the increased breast cancer incidence in Europe.

Continued expansion of the range of *M. m. domesticus* in sub-Saharan Africa.

At the time of our previous report in 2000, very little was known about the mice of sub-Saharan Africa. Breast cancer incidence was also poorly documented in this region. Recent reports have documented a near doubling in breast cancer incidence in Senegal, from 869 cases in 2012 (pop. 14.6 million) to 1817 cases in 2020 (pop. 16.7 million), exceeding the 25% increase in population (https://gco.iarc.fr/; https://borgenproject.org/6-facts-about-breast-cancer-in-senegal/, accessed on 8 March 2022). With the construction of roads inland from the coastal cities, so has the range of *M. m. domesticus* also spread inland in the past 30 years of trapping. In the decade from 1983 to 1992, *M. m. domesticus* mice were largely restricted to coastal areas, and in the north of Senegal, from 1993 to 2002, they had spread inland several hundred kilometers. In the most recent decade from 2003 to 2012, they were found right across Senegal from East to West and are now detected South of the Gambia [53]. Given that 42% of Senegal’s population lives in rural areas, it will be important to track the geographic prevalence of breast cancer incidence in Senegal with this recent expansion in the range of *M. m. domesticus*. 

*Apobec3* variant confers resistance to MMTV in mice.

One of the assumptions of the zoonosis hypothesis is that different subgenera of *Mus musculus* carry different strains of MMTV or show different susceptibilities to MMTV infection. In mice, several variants in the *Apobec3* gene (apolipoprotein B mRNA-editing complex; A3) confer resistance to infection caused by the MMTV(RIII) strain [54,55]. These variants occur in the three subgenera of the genus Mus: *M. m. musculus*, *M. m. domesticus*, *M. m. castaneus*. The distribution of *Apobec3* alleles in these mouse subgenera, and even among mouse populations of the same subgenera in different geographic locations, is highly variable and may account for differences in MMTV viral load [56]. However, other strains of MMTV have been identified in inbred laboratory mice as well as in wild-caught mice [57,58,59], and the effect of the *Apobec3* allele on susceptibility to these different MMTV strains has not been compared. We previously pointed out that *M. m. domesticus* mice have more endogenous (genome-integrated) copies of MMTV compared to *M. m. musculus* and *M. m. castaneus* mice [16], providing a wider repertoire for recombination [60] to generate different exogenous viruses. Given the diversity of MMTV exogenous viruses, some are likely to escape restriction by the Apobec3 variant. Moreover, since mouse strains lacking the *Apobec3* MMTV-resistance allele (e.g., I/LnJ, YBR and PERA mice) are still resistant to MMTV infection [55], other genetic factors also confer resistance to MMTV infection and remain to be identified. 

Human APOBEC3 variant and breast cancer susceptibility coincides with *M. m. domesticus* range.

While *Apobec3* is a single gene in mice, tandem duplications have produced a cluster of APOBEC3 (A–G) genes in humans that play a key role in protecting against the human immunodeficiency virus [61,62]. There is a common deletion in the APOBEC3 cluster that lowers the expression of APOBEC3A and APOBEC3B genes [63]. This deletion increased the risk of breast cancer in a large cohort in the United States [64] (land of *M. m. domesticus*). This finding has been replicated in a smaller study of women from Southeast Iran (land of *M. m. domesticus*). However, several other studies have not observed this association in South India [65] (land of *M. m. castaneus*), nor in large studies from Poland [63] (land of *M. m. musculus*), Norway [66] nor Sweden [67] (lands of hybrid mice between *M. m. musculus* and *M. m. domesticus*). If the deletion does contribute to breast cancer risk, the presence or absence of an association may reflect differences in endemic strains of MMTV.

Intriguingly, large studies from Malaysia [68] (land of *M. m. castaneus*) and from Shanghai (previously considered exclusive *M. m. musculus* and *M. m. castaneus* territory, see below) also found a significant association of the deletion with breast cancer risk [69]. *M. m. castaneus* mice trapped in Malaysia were reported to carry a complete endogenous MMTV proviral sequence [59], so these mice may have higher exogenous MMTV loads, whereas the recent discovery of wild-caught *M. m. domesticus* mice in Shanghai suggests a recent invasion [42] and may help explain the increased breast cancer risk associated with human APOBEC3 deletion. 

Antiretroviral therapy reduces the incidence of breast cancer.

Another human population that might be resistant to MMTV infection are individuals infected by the human immunodeficiency virus on antiretroviral therapy. Highly active antiretroviral therapy came into clinical practice in 1996; however, given that most cases would have been younger individuals, the effect of therapy on breast cancer incidence (a disease that increases in incidence in postmenopausal women in western countries) did not appear until 2010 [70]. In contrast with many other cancers, antiretroviral therapy for the treatment of HIV infection lowers the risk of breast cancer. In the first 3–5 years after HIV diagnosis, the breast cancer standardized incidence ratio (SIR) was reported at 0.6 and fell to 0.5 for the next 5–10 years after HIV diagnosis [70]. This observation has been confirmed among older persons living with HIV in the United States, for whom the SIR for breast cancer was reported at 0.61 in 2018 [71]. Whether HIV therapy prevents infection by another retrovirus (MMTV) by inhibiting its reverse transcriptase, or whether chronic immunosuppression in HIV carriers accounts for a reduced incidence, remains an open question. Antiretroviral therapy applied to the NOD.C3C4 mouse model of primary biliary cirrhosis, another human disease linked to MMTV infection [72], was shown to reduce MMTV viral loads [73].

## 5. Conclusions

The incidence of breast cancer has been growing globally. Lands that were previously free of the Western European house mouse *M. m. domesticus* have become gradually invaded (e.g., Shanghai and inland Senegal), and hybrid mice bearing exogenous strains of MMTV previously alien to these localities, may pose serious risks to the commensal human populations. If the global COVID-19 pandemic has taught us anything, it is that zoonotic infections are a serious threat to human wellbeing. Nonetheless, these epidemics can be brought under control with the rapid development and government-sponsored deployment of vaccines. A similar vaccination strategy targeting MMTV could be expected to significantly reduce breast cancer incidence and mortality worldwide.

## Figures and Tables

**Figure 1 viruses-14-00559-f001:**
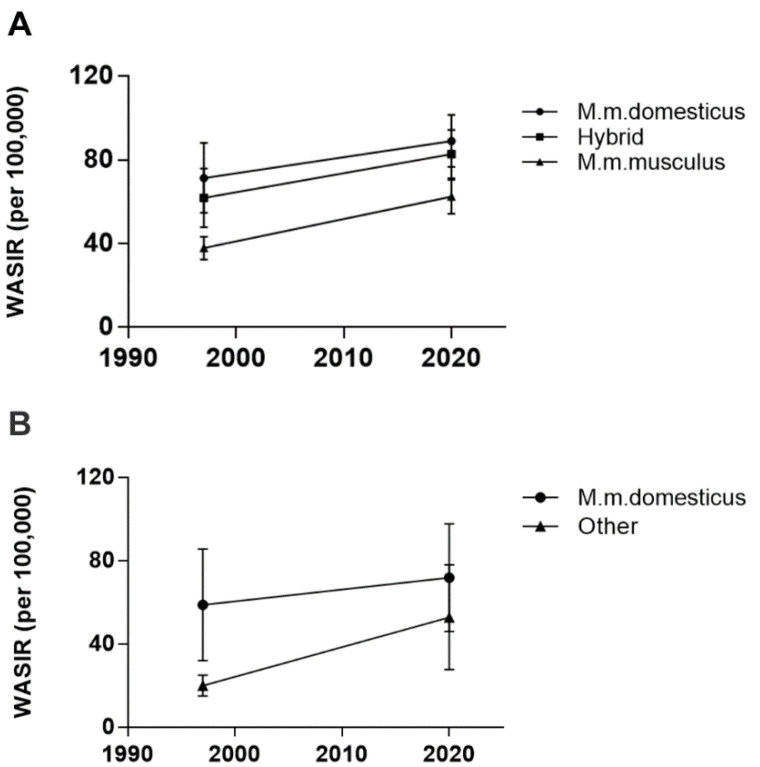
Mean world age-standardized incidence rates of breast cancer in 1997 versus 2020 according to the range of *M. m. domesticus* in Europe (**A**) and non-European lands (**B**), excluding sub-Saharan Africa. Data are presented as mean ± standard deviation.

**Figure 2 viruses-14-00559-f002:**
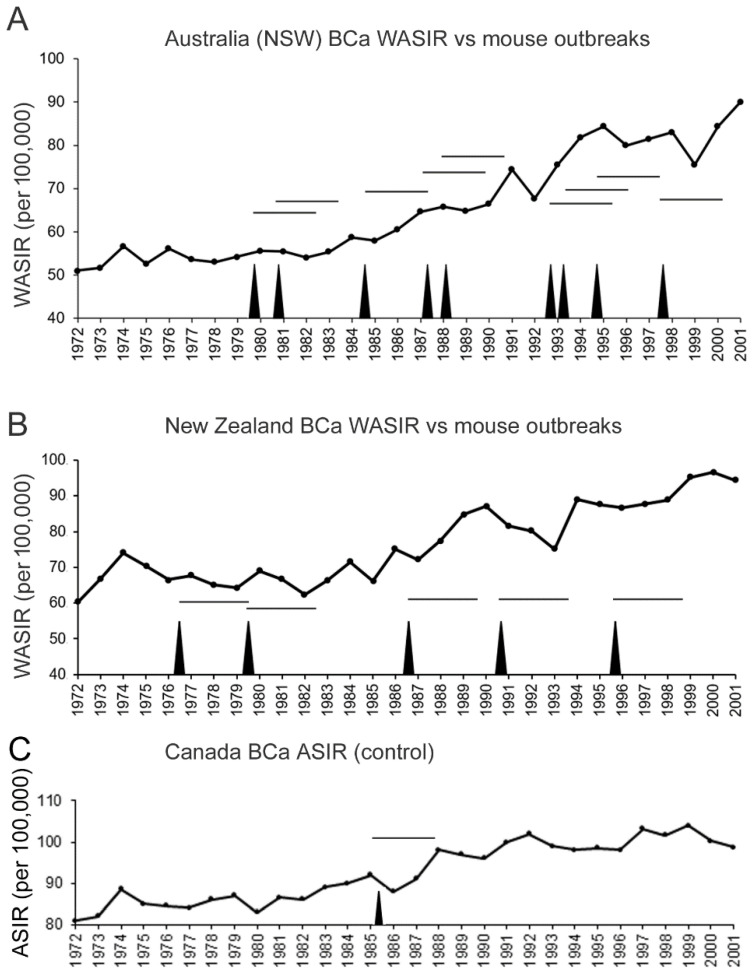
Mouse population outbreaks (black spikes) precede increases in breast cancer incidence by about 3–4 years in New South Wales (NSW), Australia (**A**) and New Zealand (**B**). World age-standardized incidence rate per 100,000 is plotted by year. Documented occurrence of mouse population outbreaks in NSW [31,32] and the Orongorongo Valley of Zealand [33] are indicated by vertical spikes. Horizontal lines show a 3-year span after an outbreak. (**C**) Canada age-standardized incidence rate per 100,000 (normalized to 1991 population) from 1972 to 2001 [2,40]. The sole vertical spike corresponds to a deer mouse population spike in the forests of Ontario [39].

**Table 1 viruses-14-00559-t001:** Change in breast cancer incidence rate from 1997 to 2020 in Europe sorted by mouse range.

		WASIR 1997	WASIR 2020
*M. m. domesticus*	Iceland	79	81
	Republic of Ireland	64	90
	UK	69	88
	Belgium	92	113
	Germany *	62	82
	France	75	99
	Spain	46	78
	Portugal	53	71
	Italy	72	87
	Netherlands	101	101
Hybrid	Norway	54	83
	Sweden	73	84
	Finland	65	92
	Denmark	73	98
	Croatia	37	69
	Austria	69	70
*M. m. musculus*	Poland	40	69
	Romania	39	66
	Hungary	NA	77
	Estonia	36	63
	Latvia	34	63
	Lithuania	29	62
	Belarus	30	52
	Ukraine	39	44
	Czech Republic	45	72
	Slovak Republic	39	60
	Slovenia	46	69

* Data for Germany 1997 are from the German province/state of “Saarland” due to privacy laws in the rest of Germany that prevented reporting of aggregate breast cancer incidence. Pairwise comparisons by Sidak’s test of the means of each group not weighted by individual populations revealed significant increases in cancer rates between 1997 and 2020 in all groups (*M. m. domesticus*, *p* = 0.0051; Hybrid, *p* = 0.0116; *M. m. musculus*, *p* = 0.0001).

**Table 2 viruses-14-00559-t002:** Change in breast cancer incidence rate from 1997 to 2020 in non-European lands (excluding sub-Saharan Africa) according to the range of *M. m. domesticus*.

		WASIR 1997	WASIR 2020
*M. m. domesticus*	Algeria	10	55.8
	Ecuador	27	38.2
	Costa Rica	29	47.5
	Peru	31	35.9
	Columbia	39	48.3
	Brazil	44	61.9
	Puerto Rico	46	68.2
	Argentina	60	73.1
	Australia	67	96
	Canada	77	82
	New Zealand	77	93
	Israel	77	78.3
	USA	79	90.3
	Hawaii	97	139
	Uruguay	93	65
Other mice	South Korea	20.8	64.2
	Thailand	12	37.8
	Taiwan	17	93
	Vietnam	18	34.2
	India	21	25.8
	China	26	39.1
	Japan	26	76.3

Pairwise comparisons by Sidak’s test of the means of each group not weighted by individual populations revealed a significant increase in cancer rates between 1997 and 2020 only for lands of Other mice (*M. m. domesticus*, *p* = 0.2501; Other, *p* = 0.0305).

## Data Availability

All data are reported within the manuscript or in supporting references.

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
