# Peer review of "Revisiting the MMTV Zoonotic Hypothesis to Account for Geographic Variation in Breast Cancer Incidence"

_viruses, 2022, doi:10.3390/v14030559_

Round 1

Reviewer 1 Report

This study belongs to a highly controversial area of research regarding the zoonosis of MMTV into the human species and its role in human breast cancer.  In particular, it presents an update on the interesting hypothesis proposed by Stewart and colleagues in 2000 that tries to explain differences in regional breast cancer rates world-wide with the host range of the M. mus domesticus species of mouse, a strain that carries the most integrated copies of MMTVs. 

Major Points:

Although the new data provided in this manuscript is interesting, it still is not sufficient to separate the role of other confounding factors that could partially explain their observations, such as the role of western diets, processed foods, and GMO crops being introduced over the years into the lands of M. mus musculus and M. mus castaneus mice.  The import of wheat from the US and other western nations into these lands is a classical example since these also not only brought the M. mus domesticus species to those lands, but also the GMO wheat.  The authors address many other factors in the manuscript, but do not mention these which are critical to their hypothesis and may also explain the 3-fold increase in breast cancer rates observed in countries with “other” mouse species.

It will be better to present the data in Tables 1 & 2 in a column graph separately for each mouse strain to allow for an improved assessment of the changes in rates observed.  Furthermore, the authors should try to show if the change in rates being observed for each country is statistically significant or not.

In terms of data presented in Figure 2, it is less convincing.  Despite having more mouse outbreaks in Australia (NSW), there are less fluctuations in the WASIR data from Australia than that from New Zealand.  Furthermore, there seems to be a linear progression in breast cancer rates between 1977 and 1990 in Australia, a time period during which five outbreaks of mice populations took place compared to three in New Zealand; yet, there is more cyclical variation apparent in the data from New Zealand.  Therefore, to strengthen this observation, the authors should show the WASIR data on breast cancer from lands of either M. mus musculus or M. mus castaneus (such as perhaps from Shenghai, China) as a control and show that the profile is without these fluctuations.  If the new data also shows the periodic fluctuations, then the result could simply be due to the other confounding factors mentioned above and not due to the spread of the M. mus domesticus species into the “other” lands.  

Since there is no definitive proof that MMTV has jumped into the human species let alone the assertion that it can cause breast cancer in humans, it is premature to propose that a vaccination strategy against MMTV is warranted in the human population similar to HPV.  Therefore, the last statement in the abstract should either be deleted or modified by adding a clause similar to the following:

“if it is proven that MMTV is an etiological factor of human breast cancer”.

Minor Points:

Line 47:  Remove “the”.

Line 90:  Rather than “promote”, don’t the authors mean “reduces” breast cancer?

Table 1:  Explain the * on Germany.

Table 2:  Remove extra space after “2020” in the title.  What are the numbers in brackets in countries with “Other” mice?  Also, there are two entries for Israel with different results with the second result showing no change.  If that is the correct data, how can that result as well as the lower rates observed in Uruguay be explained keeping the author hypothesis in mind?  Finally, it may be better to alphabetize the country lists in each category in both tables.

Line 186: Replace “would” with “may”.

Line 188: Remove “very”.

Line 189: Replace “and” with “but”.

Line 198: … occurred by “chance” not “change”.

Line 221:  Please cite references for this assertion.

Line 226: Remove extra spaces after the semi colon.

Line 267: Capitalize APOBEC3 here and everywhere else in the article.

Line 295: Remove the extra space after “in”.

Line 296: Delete “or”, add comma after [59], and replace “and” with “or”.

Line 296: Delete “polymorphism”.

Line 303: Add spaces and italicize “M. m. castaneus”.

Line 334: Replace “would be expected to” by “could”.

Reviewer 2 Report

I would like to congratulate the authors for their interest in this marginalized, yet extremely important topic. I feel that the Nobel price should be awarded to the scientists who discovered and linked MMTV to human breast cancer.

I have a few suggestions:

Line 66: Please add the mouse-dog-human hypothesis. Please add the Muchen study where breast cancer patients were more frequently dog owners.

Line 69:  please give percentages of positive breast samples, for example 20-40% in some Western Eu countries and Australia.

Line 153: Romania is twice on the list 

Round 2

Reviewer 1 Report

The authors have responded to most of the queries satisfactorily.  However, some clarifications are needed as follows:

Line 152:  Replace “than” with “as”.

Line 156:  Clarify what is “Saarland” as follows:

*Data for Germany 1997 are from the German province/state of “Saarland” due to privacy laws in the rest of Germany that prevented reporting of aggregate breast cancer incidence.

Tables 1 and 2: Add the following statement as a footnote to these tables:

Pairwise comparison of the means of each group (albeit not weighted by individual populations), revealed significant increases in cancer rates between 1997 and 2020 in all groups.

Table 2:  The clarification of the numbers in brackets is still missing.

Figure 2C:  For proper comparison and consistency, add the vertical spike at 1985 and the bar for the 3-year mark post spike in the chart.

Author Response

The authors have responded to most of the queries satisfactorily.  However, some clarifications are needed as follows:

Line 152:  Replace “than” with “as”.

Done.

Line 156:  Clarify what is “Saarland” as follows:

*Data for Germany 1997 are from the German province/state of “Saarland” due to privacy laws in the rest of Germany that prevented reporting of aggregate breast cancer incidence.

Done

Tables 1 and 2: Add the following statement as a footnote to these tables: Done

Pairwise comparison of the means of each group (albeit not weighted by individual populations), revealed significant increases in cancer rates between 1997 and 2020 in all groups.

Done.

Table 2:  The clarification of the numbers in brackets is still missing.

Apologies, numbers in brackets have been deleted.

Figure 2C:  For proper comparison and consistency, add the vertical spike at 1985 and the bar for the 3-year mark post spike in the chart.

Done.